# Direct Recycling of WC-Co Grinding Chip

**DOI:** 10.3390/ma16041347

**Published:** 2023-02-05

**Authors:** Alessio Pacini, Francesco Lupi, Andrea Rossi, Maurizia Seggiani, Michele Lanzetta

**Affiliations:** 1Department of Civil and Industrial Engineering (DICI), University of Pisa, 56126 Pisa, Italy; 2Department of Information Engineering (DII), University of Pisa, 56126 Pisa, Italy

**Keywords:** grinding, machining tools, tungsten carbide, green manufacturing, sustainability

## Abstract

Grinding is a finishing process for high precision, high surface quality parts, and hard materials, including tool fabrication and sharpening. The recycling of grinding scraps, which often contain rare and costly materials such as tungsten carbide (WC-Co), has been established for decades. However, there is a growing need for more energy-efficient and environmentally friendly recycling processes. Currently, grinding sludges, which are a mixture of abrasives, lubricants, and hard metal chips, are only treated through chemical recycling. Direct recycling (“reuse” of chips as raw material) is the most effective but not yet viable process due to the presence of contaminants. This paper presents an oil-free dry grinding process that produces high-quality chips (i.e., oil-free and with few contaminants, smaller than 60 mesh particle size) that can be directly recycled, as opposed to the oil-based wet grinding that generates sludges, which require indirect recycling. The proposed alternative recycling method is validated experimentally using WC-Co chips from a leading hard metals’ processing specialized company. The contaminant level (oxygen 0.8 wt.%, others < 0.4 wt.%), granulometry (chip D50 = 10.4 µm with grain size < 3 µm) and morphology of the recycled chips’ powder is comparable to commercial powders proving the research and industrial potential of direct recycling. The comparison of sintered products using recycled and commercial powder provided equivalent characteristics for hardness (HRA of 90.7, HV30 of 1430), porosity grade (A02-04) and grain size (<3 µm).

## 1. Introduction

In recent years, there has been a growing emphasis on promoting Sustainable Development (SD) to enable new and more innovative production models that balance economic, environmental, and social aspects [1,2]. The United Nations have defined a specific Sustainable Development Goal (i.e., SDG 12) for responsible consumption and production [1,2], and worldwide governments and organizations such as the International Organization for Standardization (ISO) have encouraged the adoption of more parsimonious production models that use less material and energy [3]. As a result, a transition from a linear “take-make-dispose” to a circular economy (CE) approach has been advocated in recent years [4,5]. Significant effort has been devoted towards recycling a variety of materials, including the recovery of valuable metals from mining [6], circuit boards [7] or batteries [8], the recycling of aggregates such as concrete [9,10] and ceramic waste [11]; polymeric waste [12]; glass waste [13]; and other hybrid materials such as carbon fiber-reinforced composites [14]. One of the most used frameworks for transitioning to more sustainable products, processes, and systems, is the 6Rs model, which is depicted in Figure 1 [5].

This work focuses on two of the top 30 rare raw materials according to the 2020 European Union (EU) list [15], Tungsten (W) and Cobalt (Co), which are the main components of tools and inserts used in machining. Manufacturers of rotary tools (such as drills and mills) start from cemented carbide bars. The highest quality tools rotating at high speed require high accuracy, and low roughness, cemented carbide bars. Finishing is achieved by the centerless grinding of bars from sintered raw powder, such as tungsten carbide (WC-Co) also known as hard metal, generating chip scraps that can be recycled. New powder mixtures, including recycled chips and powders, are receiving attention from various fast-growing Additive Manufacturing (AM) processes, extending even further the potential applications of recycling powders [16].

As illustrated in Figure 1, circularity can be achieved by *rethinking* the grinding process and *reducing* contaminants in the output scrap (e.g., chips), thereby enabling novel recycling routes based on recycled WC-Co powder, and promoting a special waste to secondary material, containing two critical raw materials. This represents the primary contribution of this work.

### 1.1. WC-Co Raw Materials

A liquid phase-sintering process is used to produce WC-Co, where the hard tungsten monocarbide (WC) grain particles are bonded together in a tough binder matrix (e.g., Cobalt (Co)) [17]. The combination of WC grains and metallic Co binder exhibits an excellent sintering behavior, leading to outstanding densification (pore-free structure) at the time of liquid phase sintering. Properties such as high compressive strength, hardness, toughness, and especially red hardness properties (i.e., retains its hardness at high temperatures) [18] are some of the fundamental WC-Co properties to withstand various forms of wear, corrosion, and high levels of compressive stress [19].

In the last half century, and especially in the current geopolitical framework, new challenges have emerged in relation to the supply and demand of natural resources and raw materials [15,20,21], and there is an urgent need to improve the circularity of critical raw materials [22]. Due to its wide range of applications and high usage, WC-Co faces many challenges related to the tungsten and cobalt elements, which are strategic for the functioning of modern technologies, economies, or national security (i.e., economic impact), and critically subject to the high risk of supply chains’ disruption (i.e., supply risk) [23]. These materials have been identified as “critical raw materials” by the EU since 2011 [15] and as “critical minerals” by the United States (US) Department of Interior in 2018 [24]. Other governments, such as Australia [25] and Brazil [26], have also recently issued a list of specific strategic minerals and metals including W and Co. China’s first official catalogue of “strategic minerals” was established by the National Plan for Mineral Resources (NPMR 2016–2020) and also includes W and Co [23].

Tungsten is a rare hard metal that is primarily derived from natural resources and has low global abundance [27]. The most significant and abundant tungsten deposits have been found in Southeast China (with a dominant production over 80%) and Southeast Asia (e.g., Vietnam), followed by Canada and Russia. Furthermore, substantial tungsten resources are located in the Central Andean belt, the East Australian belt, the Mesoproterozoic Karagwe–Ankole belt, and the European Variscan region [28]. It may be surprising to some that the rare earth element (REE) and tungsten are considered “strategic minerals” as China is the leading producer of these materials. The reason they are deemed “critical” by the US, EU, and others is because of concerns over supply security due to China’s dominance in the supply chain of these materials [23].

The chief primary sources of tungsten in the world, such as Scheelite and Wolframite, contain 0.1–5% WO_3_ and require the mining of a huge ore tonnage to gain a small amount of tungsten [27].

Tungsten’s most important use today is the production of tungsten carbide as the main component (40–95 wt.%) in the form of tungsten monocarbide (WC), which has a hardness close to that of diamond (8–9 of 10, on the Mohs scale) [29]. In addition to hard metal industries and cemented carbide tool-bits [30], tungsten as a metal is also used for developing super alloys (e.g., turbine blades, jet nozzles, or other parts of jet engines) [31] and deposited as special coating to increase wear and corrosion resistance [32].

If China possesses the world’s widest reserves of tungsten, the Democratic Republic of Congo (DCR) owns the dominant reserves of cobalt (about 50% of the world’s cobalt production is mined in DCR) [33,34]. Australia and Cuba follow, making up around 20% and 10%, respectively, while minor reserves can be found in the Philippines, Russia, Canada, Madagascar, and Zambia [34].

Cobalt is the most compatible binder for cemented carbides [35]: mixed with WC, it creates a eutectic compound that lowers the melting temperature and thus the processing temperature (more than 2800 °C for pure WC versus 1300–1500 °C with adequate Co additions) [36]. Moreover, the wettability of the carbide phase by Co is excellent, and WC is easily soluble in cobalt [30]. The conventional concentrations of Co for diverse applications range from 3 to 15% [27], the highest concentrations being undesirable for health reasons. Please note that the hardness depends on the cobalt content and the grain size of the carbides [37]. Unfortunately, the price of Co has increased significantly due to the fast growth in demand for the batteries of electric vehicles [34,38]. Nickel, iron, and copper have been proposed as alternatives to cobalt, but they are less efficient binders [39,40].

### 1.2. WC-Co Scraps and Recycling

As discussed in [18,41], the extraction of W and Co metals from their ores is more demanding than from recycling scraps. For example, scrap materials typically contain higher concentrations of tungsten, with levels ranging from 10 to 75 (wt.%) of WO_3_. Since the third decade of the 90s, increasing attention has been dedicated to recycling techniques of tungsten, handling this relatively rare and expensive commodity with great care, trying to use it efficiently, and not spoiling materials that could be reprocessed as valuable secondary resources.

In this context, the EU concept of the circular economy [42] has been recently proposed as a promising solution to address the limitations caused by the depletion of natural resources and promoting SD [43]. According to previous studies [27,43], recycling is a crucial component of the global tungsten flow, with secondary tungsten resources contributing to roughly 34% of the global tungsten consumption [43]. This includes 10% from scrap generated during processing and approximately 24% from used or worn-out tungsten carbide parts from industry [44].

Many tungsten recycling technologies have so far been developed and industrially adopted. These technologies can be broadly classified into two main categories based on the level of the chemical-intensive process used [43]:-*direct* recycling: scraps are transformed into powders with the same composition. In direct recycling, scrap materials are directly transformed into a product having a similar composition, using physical or low-intensive chemical processing methods (or a combination of both [45,46]). High-purity scraps (i.e., the same composition as the final product and precise size-range) must be sorted and *cleaned* during the manufacturing process to be further processed. *Direct* recycling is combined with a minimum of energy consumption, chemical waste generation, and production cost. For those tungsten carbide scrap materials that fail to meet the strict purity standards required by direct processes [27], chemical methods are needed;-*chemical* recycling methods include both *indirect* and *semi-indirect* processes. *Indirect* methods involve the conversion of scraps into intermediate products, such as Ammonium Para Tungstate (APT), while *semi-indirect* methods involve the selective dissolution of one component, typically the binder. These methods require the use of intensive chemical modification techniques, such as the use of acid or alkaline media, to obtain intermediate products that can be further processed. For the sake of readability in this article, we will consider chemical recycling the same as indirect.

Each recycling technology, whether more or less chemically intensive, older or newer, and whether covering a larger or smaller share of tonnage, has been designed with a specific type of scrap in mind, from end-of-life products to swarf/turning/sludges generated during machining [36].

The WC-Co grinding sludge, which is a mixture of abrasives, lubricants, and hard metal chips, often requires chemical recycling due to the presence of oil, refrigerants, and other impurities or contaminants.

The current study aims to evaluate the feasibility of the direct recycling of WC-Co chips generated in an industrial oil-free dry grinding process, which yields the highest purity WC-Co chips from dried sludge.

The remainder of this paper is organized as follows. The state of the art of the WC-Co grinding process and an overview of various sludge recycling methods are presented in Section 2. The analysis of the WC-Co chips generated in an industrial oil-free dry grinding process is described in Section 3. The proposed direct recycling process for the chips analyzed in Section 3 is outlined in Section 4. Finally, Section 5 and Section 6 discuss the proposed recycling process performance and limitations along with conclusions and existing challenges.

## 2. State of the Art of WC-Co Grinding Sludge Recycling Methods

Section 2.1 provides a background on the WC-Co lifecycle and processing, from cradle (i.e., mineral mining) to gate (i.e., the semifinished or finished WC-Co product) introducing the reader to the main general background in the field. Section 2.2 offers a detailed overview of the different recycling strategies (i.e., direct, and indirect) for WC-Co grinding sludges. Section 2.3 outlines the major impurity sources of grinded sludge preventing direct recycling.

### 2.1. WC-Co Lifecycle and Processing Overview

The flowchart of Figure 2 provides the WC-Co lifecycle and processing overview, from natural resources mining to WC-Co sintered products’ final usage, including the conventional recycling routes, according to the authors of [47].

-*Mining*. The first step is the extraction of natural resources through drilling, blasting, and digging. The ore is then crushed and milled [31]. The primary resources, such as wolframite (Fe,Mn)WO_4_ and scheelite CaWO_4_, are extracted and transported and then processed further through hydrometallurgy.-*Hydrometallurgy*. This step includes sodium hydroxide NaOH and sodium carbonate Na_2_CO_3_ digestion [48], which, after filtration and precipitation to remove silica and molybdenum impurities [49], leads to pure sodium tungstate Na_2_WO_4_. Tungsten is extracted from the purified Na_2_WO_4_ solution using an organic solvent mixed with sulfuric acid H_2_SO_4_. The extract is washed with deionized H_2_O and then the organic solvent is stripped from the tungsten by the addition of NH_3_ [49], resulting in an ammonia tungstate ((NH_4_)_2_WO_4_) solution, which is treated to form APT [49,50].-*Pyrometallurgy*. At the third step, the APT is transformed to tungsten blue oxide (TBO) WO_x_ via calcination in rotary furnaces [51]. Then, hydrogen H_2_ reduction converts TBO to tungsten metal powder in push-type furnaces with stoichiometric H_2_ excess [49]. Finally, carburization is commonly performed by introducing tungsten metal powder blended with carbon black in a furnace with a reducing H_2_ atmosphere [49] to obtain WC powder.-*Powder metallurgy*. WC and Co powders are mixed in variable shares based on the application, along with a solvent (e.g., hydrocarbons [52]) and additives (e.g., paraffin wax, 1–3% share of the mixture). The mixture is then milled to obtain the desired particle size [53,54] and granulation is typically carried out via spray drying [53]. Finally, the ready-to-press powder is compacted into the desired shape (e.g., cylindrical) and sintered until 1450 °C in an inert atmosphere to avoid oxidation.-*Machining* (*cutting*, *grinding*, *milling*). Material-removal operations are carried out to cut the cylindrical items, reduce the surface roughness as well as geometrical errors (e.g., run-out) within the required tolerance [55], and reach the final shape of the cutting edges (e.g., helicoidal). As shown in Figure 2, different sources of WC-Co waste are generated during all these subtractive machining steps, including solid swarf and chips (i.e., hard scrap), and very fine chips mixed with fluids and contaminants (i.e., sludges) [56].-*Use*. Once the finished cylindrical product is used, at the end-of-life, it must be disposed of and can also enter the recycling route [56].

Without loss of generality and with the many potential extensions presented, the direct recycling method proposed considers as input the chips generated via the oil-free dry grinding of WC-Co cylindrical bars (Figure 2, yellow box) for high-speed, high-accuracy machining tools.

### 2.2. WC-Co Grinding Sludge Recycling

As previously introduced in Section 1.2, several methods for recycling tungsten have been developed in response to the large variety of scraps available in industry [56]. Figure 2, upper side (orange and blue dotted lines) provides a graphical overview of the two conventional recycling routes for WC-Co scraps (i.e., *direct* and *indirect or chemical recycling*), covering about 50% of the total tungsten recycling share [31].

According to the literature, and as graphically shown in Figure 2 (blue dotted lines), direct recycling routes exhibit low economic costs and reduced environmental impact compared to chemical recycling (Figure 2, orange dotted lines). However, conventional direct recycling processes, such as Zn, Coldstream, oxidation-reduction, bloating–crushing [57], are not suitable for grinding sludge generated by oil-based wet grinding. This is because of the tight constraints in the input material composition [57]. As a result, the severe variation of the Co, Ni, Cr, Fe rates which can depend on the variable batch production and the presence of contaminants such as refrigerant fluids, prevents the use of any direct approach for WC-Co sludges. In a recent review on grinding tools [16], recycling methods involving the magnetic separation of metal chips from sludge and chemicals’ treatment in the presence of oil were discussed, but the case of carbides was not considered [16].

On the other end, the chemical modification process of WC-Co sludges involves similar steps to the traditional powder production from concentrates. Unlike direct recycling, which requires pure and sorted WC-Co scrap, chemical recycling accepts a broad range of scraps, even with a high impurity level and varying compositions, such as cutting or grinding sludge (Figure 2; orange dotted lines). In chemical recycling, intermediate products, such as APT, are created from all types of scrap and then processed, yielding outputs of equivalent purity as the virgin material. This allows for the integration of primary (i.e., mined) and secondary (i.e., scraps) raw materials in production lines and the recycling of large volumes [43]. Despite its costs (in the order of tungsten W-concentrates conversion) and limitations, such as prolonged conversion rates, substantial waste by-products, and high consumption of energy and reagents [18], the chemical recycling of WC-Co scrap remains the standard de facto method for recycling grinding sludges [58].

### 2.3. WC-Co Grinding Sludge Impurity Sources

Given the limitations of direct recycling, achieving optimal recycling at the highest utilization level will necessitate taking scrap purity into account. This also presents a future opportunity to investigate alternative manufacturing processes that can handle the direct recycling of grinding sludge, which is currently considered impure scrap and chemically treated [27].

The variables impacting the WC-Co sludge purity in grinding can be systematically analyzed using the Ishikawa or fishbone diagram also known as the 4Ms (Man, Machine, Method, Material) [59], as shown in Figure 3. In the following, a short description of each variable will be provided.

-*Machine*: Cemented carbide cylinders are machined using various grinding machines and conditions (such as dry and wet) [16,60]. Conventional grinding often employs a wheel with diamond abrasive grains of varying sizes, ranging from hundreds (roughing) to a few (polishing) μm, which are embedded in a metal or resinoid binder [16]. In combination, a water–oil emulsion is used as a coolant during the grinding process to prevent burnout and thermal damage, lubricate the tools, reduce wheel wear, and remove grits from the grinder [61]. Among several types, there are two significant categories of water-based grinding fluids: mineral oil in water emulsion and semi/synthetic products in water coolants. Both are mixtures of oil and synthetic lubricant with an average dilution rate of 1–5%. Still, considering the machine variables, the major process parameters, such as table speed, abrasive material, grain size, wheel material, diameter, speed, and depth of cut, have an enormous importance in the final product quality as well as the generated sludge purity level.-*Methods*: As another variable, quality management methods or procedures greatly influence the sludge’s characteristics. Procedures for the input material control (i.e., incoming quality control), determine the average chemical composition of the input material before the grinding process. Additionally, specific grinding-process control (e.g., process parameters monitoring via sensing techniques), machine and tools monitoring strategy (e.g., wear control), as well as handling and storage, determine the final physical and chemical characteristics of the WC-Co sludge, including the level of organic contaminants and oxidation. Furthermore, production planning and sludge storage affect the level of variation in the output sludge from batch production.-*Material*: The characteristics of the output WC-Co sludge will vary depending on the composition of the input WC-Co raw material.-*Man*: Based on the operators’ skills, experiences, and training levels as well as according to the previously defined procedure to be followed, the WC-Co sludge presents specific characteristics.

This study focuses on the direct recycling approach developed to handle a specific sludge generated by a patented centerless oil-free dry grinding machine and methodology [62]. This machine and process produce purer sludge compared to conventional grinding approaches, which is oil-free chips, with lower oxidation levels (due to low grinding temperature) and contaminants’ levels (e.g., Ni, Fe, Mo, Cu) within ranges. Section 3 presents a characterization of the chips’ output from the oil-free dry grinding and describes the developed direct recycling strategy.

## 3. Grinding Chip Characterization

In this section, we present the methodology used to analyze the chips obtained from a patented industrial machine and methods for oil-free dry-grinding hard metals [62]. Samples of chips have been collected from various lots produced across several years (2010–2022).

The chips’ analysis has not been straightforward and has required using, validating, and integrating different laboratory tools and techniques. Among the main difficulties are:-The specific composition of the material and its particle size (fine chips) does not allow for quick X-Ray Fluorescence (XRF) analysis, as matrices and calibration standards for XRF are not readily available.-The small particle size of the material causes large peaks when using X-Ray Diffraction (XRD), making it difficult to reliably read the spectra as the peaks of Co and W (and of the respective oxides) may overlap.-The small particle size and tendency to aggregate negatively impact the success of a particle size analysis by laser diffraction techniques (i.e., granulometry).

### 3.1. Chip Morphology, Granulometry and Composition

The input material for this study consists of brittle and easily grindable chip. Some images obtained with a scanning electron microscope (SEM) (FEI Quanta 450 FEG ESEM instrument (Hillsboro, OR, USA)) at different magnifications are shown in Figure 4a,b. According to the analysis carried out using Helos (H4286) and Oasisdry/L R2 laser diffractor (Figure 5), the average sample particle size can be roughly classified into three average values: D10 = 3.4 µm, D50 = 10.4 µm, D90 = 25.3 µm. The analysis has been validated suspending the samples in water medium after stirring for 1 h via ultrasound and analyzing by additional laser diffractor instruments (i.e., Bettersizer S3 Plus Bettersize, Bettersize Instruments Ltd., Dandong, China and Mastersizer 3000 Malvern, Malvern Panalytical Ltd., Malvern, UK).

Please note that the SEM images (Figure 4a,b) show chips bigger than 50 µm that are not captured during the laser diffraction analysis (Figure 5) due to the fragility of the chips themselves, which tend to break during the analysis. It is important to note that the particle size analysis should not be confused with the WC grain size (main parameter affecting the mechanical properties of the product and its commercial value [63]). The values obtained for the particle size analysis refer to the size of the chips (i.e., aggregates of WC grains held together by Co as shown in Figure 4b). The grain size is measured by crystallographic analysis on a sintered sample made with input chips in their current state, results in less than 3 µm maximum grains size, as shown in Figure 4c.

Given the input chips’ fine granulometry (<74 µm), the chips can be easily processed to reduce and make uniform the granularity with conventional ball milling and obtain a WC-Co powder [43].

From a composition perspective, the different samples of input material (collected over years) presented a similar percentage of Co and a similar level of contamination (the absolute values are confidential). In this case, we focused on identifying the critical contaminants (also known as impurities) to develop a recovery strategy. The characterization of the chip samples has allowed for the evaluation of industrially applicable recycling alternatives.

The absence of lubricant oil and the limited content of contaminants from wear of grinding wheels of the patented machine [62] differentiate the input material (i.e., dry chips) from traditional grinding sludges for which there is not yet a direct recycling method.

### 3.2. Comparison with Commercial Powders

The goal of this work is to directly recycle the oil-free dry ground chips to obtain a powder comparable to the commercial one that can be used for the sintering of new hard metal products.

While a detailed and recent review focusing on numerous types of hard metal microstructures has been published [64], a comprehensive up-to-date review for WC-Co chemical composition is not available. As generally remarked by numerous scientific works, the most common chemical composition for WC-Co hard metals is typically a mixture of tungsten carbide and cobalt in the amount of around 85–95% WC and 5–15% Co in addition to other minor components highly dependent on the given application field [37].

Regarding the definition of a benchmark, to our knowledge an official reference does not seem to be available in the literature, instead it varies over time. With the advancement of technology and manufacturing processes, commercially available WC-Co powders present more consistent and uniform particle sizes and compositions. For this reason, specific vendor datasheets from reference websites are considered here as a benchmark [65]. In addition, a scientific publication [66] and an EU official report [67] on impurities is used for defining the maximum acceptable levels of contaminants.

In the following Section 4, the novel recycling method based on the chips generated by the oil-free dry grinding machine is detailed.

## 4. The Proposed Direct Recycling Method

In Section 3, the material composition and grain size was compared with a benchmark commercial powder [65] showing overall acceptable contaminants’ levels, except for the oxygen (O) content (due to oxidation of the material) that is above the limits and the carbon (C) that is not in the carbon content range. This does not allow the use of the input material in its current state. On the other hand, the WC grain size under 3 µm and the fragility of the chips allow for grinding. This section outlines the proposed direct recycling method and its validation (next section) at a laboratory scale. Figure 6 summarizes the main steps of the proposed method, and also includes a comparison with the chemical approach currently used for conventional wet-grinding sludges. The main steps are:*Pre grinding*: remove possible sources of superficial contaminants through a simple cleaning of the sintered bar surface. This step is useful to easily remove contaminants such as C, Ca, and others that have been observed and may be present on the bars’ surfaces before grinding.*Grinding*: conduct an oil-free dry grinding using a centerless grinding machine with segmented wheels [16,62] to produce high-purity and fine (smaller than 60 mesh) chips.*Post grinding*: reduce the total internal oxygen content of the ground chips to avoid defective power, which in turn leads to a subsequent defective sintered product [68]. Additionally, the content of C must be carefully checked to avoid the formation of a graphite or η-phase in the sintered part (brittle) [69]. In this step, we have compared two alternative methods: treatment in H_2_ up to 400 °C [70] and treatment in Argon (Ar) up to 900 °C [68]. The former approach (H_2_) requires longer times but does not modify the C content (which can subsequently be set by adding virgin or recycled W, carbon black, Co powders). The latter approach (Ar) presents faster and more efficiently replicable characteristics at a laboratory scale, but it does not allow precise control of the C content and does not prevent decarburization of the WC.*Chip milling*: reduce the granulometry of the output chips to create a fine powder and mix the virgin W, carbon black, or Co powders to achieve specific compositions, if necessary [54]. The powder has been milled in an MGS Mills planetarium ball mill (440 rpm) for 30 min, loaded with WC-Co balls (3–6 mm) and distilled water as a milling medium. Correction of W, C, and Co can be applied at this stage to improve homogeneity. After milling/mixing, conventional granulation and sieving can generate ready-to-press powder.

## 5. Direct Recycling Method Validation

Table 1 reports the material composition before and after the proposed recycling process. The level of contaminants present in the recycled powder is also compared to the target level present in typical virgin powders retrieved from the literature [66,67] and merged with the commercial data for Zn recycled powders [65]. As shown, the post-grinding step (i.e., H_2_-N_2_ treatment) does not modify the C content if under 500 °C, which can be further corrected. A significant reduction in O from 3.6% to 0.8% is observed, but the H_2_-N_2_ treatment has not completely removed the oxides of the residual W. The 0.8% O can be removed during the end phases of the sintering process.

The same granulometry method applied to chip particles (Section 3, Figure 5) has been used to test the granulometry of the powder. Resulting values D10 = 0.6 µm, D50 = 2.1 µm, D90 = 6.8 µm show that the average particle size has been significantly reduced compared to the input chips (Section 3) and meets the commercial values [65].

Similarly, to Section 3, the WC grains embedded in the Co binder of the chips present a size lower than 3 µm, similarly to the grain size of the commercial powder, ranging from 10 µm to sub-micrometer. SEM images, obtained using a Hitachi TM3030Plus, of the starting chips and recycled powder are reported in Figure 7a,b, respectively.

The industrial use of the recycled powder has been tested in comparison to a commercial powder, via conventional sintering of cylindrical bars Ø 6 mm. The sintered material, obtained with the powder (Table 1) recycled with the proposed direct approach (Section 4), presents comparable properties to WC-Co bars obtained with commercial powders having the same Co content. Table 2 summarizes the main properties of the sintered, recycled powder compared to the commercial “virgin” powder, using two different batches of sintered bars.

## 6. Discussion

The oil-free dry ground chips have shown clear benefits over grinding sludge from oil-based wet grinding.

Comparing the composition of the input chips with the contaminants’ limits (Table 1) yields:-Cr, V, Nb, and Ta, commonly used as grain growth inhibitors (desired values are defined in [77]), are present in tolerable percentages.-Fe, Ni, Cr, Mo, and V, deriving from the wear of the steel mills [67], are within limits.-The oxygen (O) content (due to partial oxidation of the material in air) is above the limits preventing the use of the input material in its current state. The carbon C content also does not fit into the project carbon content range [69].-Other critical elements are Al, Si, Cu, Ca, and S.-Cu, which is not usually present in virgin powders, is present in high percentages.

The small differences of the main contaminants in the two samples (Table 1) may have occurred due to the series of processes: pre grinding, grinding, post grinding, mixing, and virgin powder dilution.

Despite the fact that the content of some contaminants is greater than that declared in commercial powders, it may certainly be acceptable for many applications.

Pre-grinding material cleaning, post powder heat-treatment, milling, and mixing virgin powder are key actions to control the output composition and obtain a new secondary material enabling circularity (Figure 8; light green boxes). This aspect is further detailed in Section 6.1. Future directions are outlined in Section 6.2.

### 6.1. Sustainability of the Direct Recycling Method

It has been shown that the powder from the recycled chips can be directly used for sintering new hard metal parts, closing the loop for a novel sustainable development model based on raw material circularity (Figure 8).

Table 2 shows that the grain size and hardness properties are comparable. The porosity of the recycled powder after sintering bars is bigger but still in line with the higher standard requirements [76].

The main limitation intrinsic to the circularity of the proposed method is generally found in direct methods: the increase of contaminants over loops. As graphically shown in Figure 8, the inclusion of virgin powder (e.g., virgin cobalt or virgin tungsten carbide) to the recycled powder serves to mitigate the accumulation of impurities, such as copper, aluminum oxide resulting from grinder tool wear, or iron, nickel, chromium, molybdenum, and vanadium resulting from mill wear, beyond acceptable levels, even after multiple recycling cycles [66]. This is crucial to ensure that the quality of the final product is not negatively impacted. This can be achieved through regular monitoring of the composition of scrap chips and by knowing that the virgin powder may also contain impurities (detailed in the left-hand side of Figure 8), which must be considered during the addition of virgin powder to achieve the desired composition (e.g., 70/30 wt.%/wt.% as in the present case). Mixing the recycled powder with virgin powders extends the circularity over repetitive loops.

Among possible solutions and approaches in addition to mixing are: (i) reset the sludge via indirect chemical processing after a threshold on specific contaminants is over the control limit; (ii) extend the root-cause analysis (RCA) to the whole production process and plant to identify the critical activities or variables impacting the level of contamination and propose corrective actions, e.g., according to the Ishikawa diagram (Section 2), in the same way as it has been executed in this work to identify the C, Cu, and Al_2_O_3_ contaminants coming from the grinding wheel wear, additional (confidential) contaminants used during sintering, and Fe, Ni, Cr, V, and Mo coming from the mill wear.

### 6.2. Future Directions

These promising results address new research directions:-Various stakeholders, including manufacturers, users, and recyclers, can be identified from a research and industrial perspective with long-term, well-defined relationships between the value chain actors that may cover the separate activities shown in Figure 8. Such relationships can be enhanced by vertical integration, or by long-term contractual agreements [22].-Scaling up the proposed methodology for the direct recycling of WC-Co chips from laboratory to industrial context. This analysis entails a disruptive change in the business model of WC-Co bars production and finishing for greener manufacturing [78].-Exploring additive manufacturing and other powder metallurgy processes using the directly recycled WC-Co powder in the interest of energy, resources, and sustainable development of alternative sintering methods. WC-Co hard metal parts are usually produced by casting and powder metallurgy, which are limited by complex geometries and require post-processing such as conventional and non-conventional machining [37,79].

## 7. Conclusions

The recycling process of carbide-manufactured items (WC-Co) is a crucial stage in the tungsten supply chain. While there are various recovery methods available, each designed for a specific type of scrap, none of these techniques allows for the direct, non-chemical recycling of grinding sludge and chip, as proposed in this study. Through chemical, morphological, and other laboratory analyses, the properties of the input material (chips) have been evaluated to determine their proximity to commercially recycled powders, in particular the concentration of O. The morphological characteristics of the chips highlight a fragile structure that is easily grindable. Additionally, the grain sizes of the WC in the material have been found to be below 3 µm, indicating a promising aspect for further processing. In the subsequent investigation, a recycling method has been defined to produce recycled powder starting from chips and the same analysis has been repeated post-treatment. The use of an oil-free dry grinding process, in conjunction with preceding and subsequent cleaning stages and final milling, has allowed for the production of secondary materials and sintered products characterized by low contaminant levels comparable to commercial powders in terms of composition, granulometry, and morphology. From a business-model perspective, the circularity of WC-Co is a fundamental enabler for a successful raw materials recycling-strategy.

## Figures and Tables

**Figure 1 materials-16-01347-f001:**
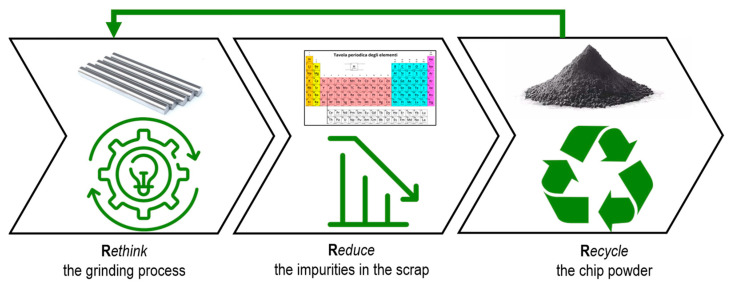
Three of the 6Rs method (i.e., *rethink*, *reduce*, *repair*, *reuse*, *refuse*, *recycle*) considered in the current work for the recycling of WC-Co ground chip.

**Figure 2 materials-16-01347-f002:**
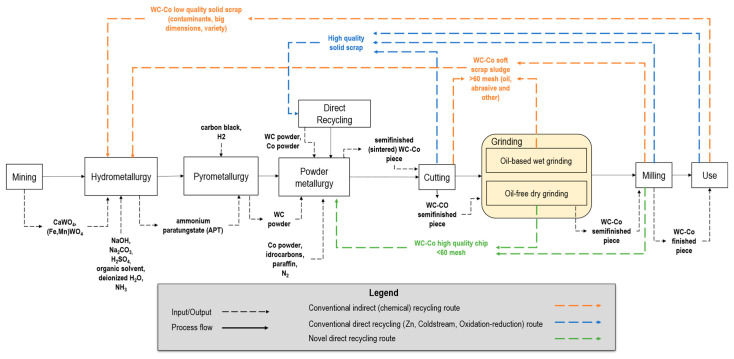
Flowchart for the typical WC-Co production process till the final use and recycling routes, adapted from [47]. The grinding step is highlighted due to the potential for triggering novel direct recycling routes (green dotted line) as addressed in the current work.

**Figure 3 materials-16-01347-f003:**
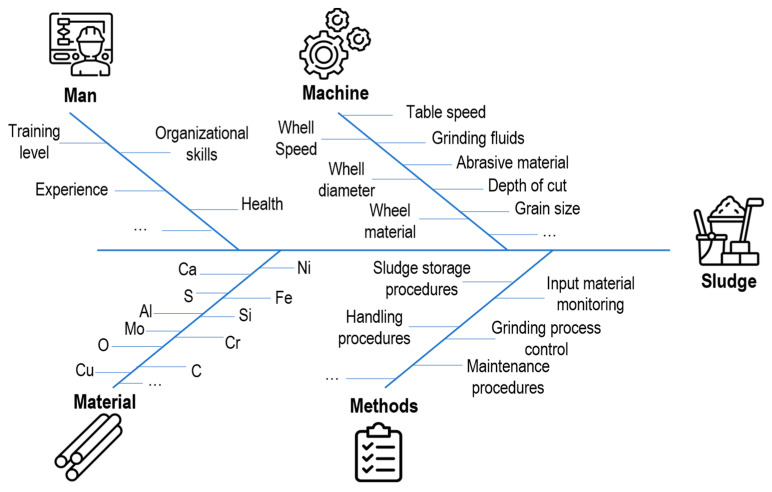
Ishikawa (fishbone, 4Ms, or cause–effect) diagram highlighting the main identified variables and factors impacting the WC-Co sludge conventional grinding wheel.

**Figure 4 materials-16-01347-f004:**
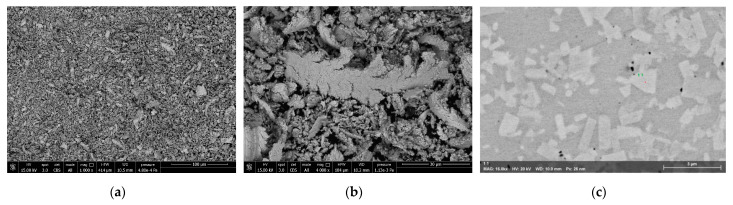
Scanning electron microscopy image of a sample taken by SEM Quanta 450 FEG FEI Company, showing (**a**) the initial chips, (**b**) a magnified view of the shape, and (**c**) the chips after sintering process.

**Figure 5 materials-16-01347-f005:**
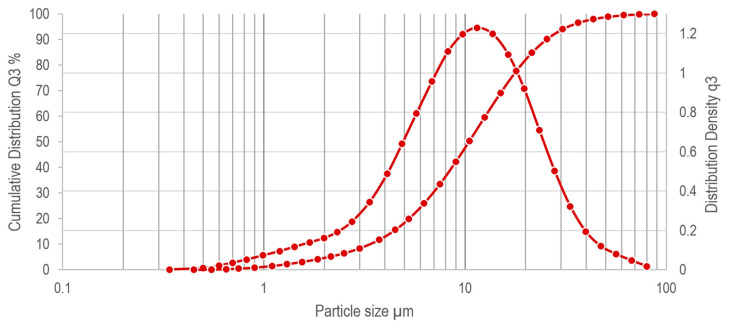
Helos (H4286) and Oasisdry/L R2 particle size analysis averaged over two input chip samples using the following parameters. Software: PAQXOS 5.0, Pressure: 1 bar, Vacuum: 46 mbar, Start: Copt > 2.1%, Valid: Always, Stop: 10 s Copt < 1.9% or 99 s real time, Feeder: VIBRI; Feed rate: 68%; Gap width: 4 mm.

**Figure 6 materials-16-01347-f006:**
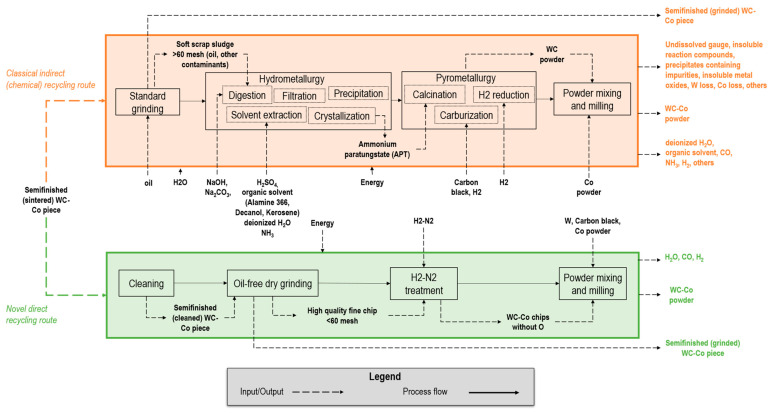
Flowchart for the conventional (orange box, upper side) and unconventional (“green” box, lower side) grinding scrap recycling starting from the same semi-finished WC-Co piece.

**Figure 7 materials-16-01347-f007:**
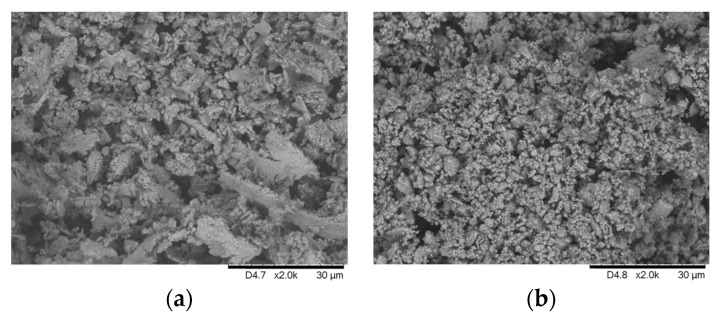
SEM TM3030Plus, Hitachi images showing the crushing of the input chips (**a**) to obtain the recycled powder (**b**).

**Figure 8 materials-16-01347-f008:**
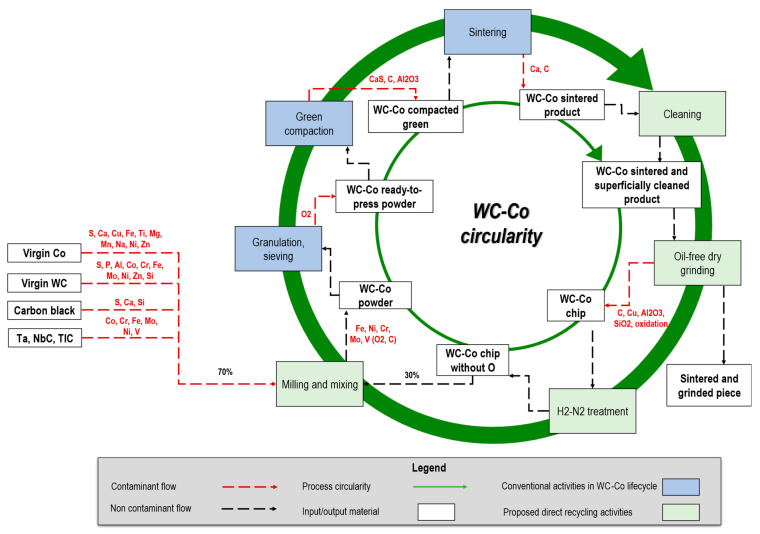
The circularity of the proposed direct recycling approach with identification of the sources of contaminants. Mixing the recycled powder with virgin powders extends the circularity over repetitive loops.

**Table 1 materials-16-01347-t001:** Content (wt.%) of main contaminants of the WC-Co material (excluding W and Co) in the input chip, obtained via oil-free dry grinding, and in the recycled powder obtained following the recycling method proposed in Section 4 (the reported values are the average of two repetitions). Last column provides the upper limits on the main contaminants according to [65,66,67].

Main Contaminants	Input Chips	Recycled Powder	Upper Limit [65,66,67]
C	5.92	5.83	Carbon content range ^1^
O	3.60	0.80	0.50
Cr	0.46	0.41	0.60
Al	0.10	0.25	0.05
Si	0.10	0.15	0.10
Mo	0.10	0.03	0.10
V	0.09	0.11	0.20
Ni	0.16	0.08	0.50
Fe	0.06	0.12	0.30
Cu	0.02	0.02	0.03
Ca	0.06	0.04	0.01
S	0.06	0.05	0.01

^1^ whose amplitude decreases with decreasing Co content. Higher Co content limits unwanted compounds formation even without exact control of the C content [69].

**Table 2 materials-16-01347-t002:** Main properties of two lots of sintered parts using as input material the recycled powder and commercial “virgin” powder [65]. The main properties from the industrial application viewpoint and the related reference standard used for the measurements are reported in the first two columns.

Properties	Reference ISO Standard	Recycled Powder Sintered	Commercial “Virgin” Powder Sintered [65]
Density (g/cm^3^)	ISO 3369 [71]	14	14.4
Grain size (µm)	ISO 4499-2 [72]	3 (medium)	1–3 (medium-fine)
Hardness (HRA)	ISO 3738-1/2 [73,74]	90.7	90.7
Hardness (HV30)	ISO 6507-1 [75]	1430	1435
Porosity (grade A-B-C)	ISO4499-4 [76]	A04-06	A02

## Data Availability

Not applicable.

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
