# Peer review of "Direct Recycling of WC-Co Grinding Chip"

_materials, 2023, doi:10.3390/ma16041347_

Round 1

Reviewer 1 Report

Comments to the authors

Manuscript materials-2133361

Title: Direct recycling of WC-Co grinding chip

General comments:

The study aimed the direct recycling of tungsten carbide with an oil-free dry grinding process. The manuscript is well-written and presents the main subjects of this topic. Still, there are grammar and English mistakes to be corrected. A careful revision is necessary.

Is it a review article? Because there are more literature review than results.

-       I recommend the authors to cite different fronts on SDGs, such as recycling of urban solid waste (https://doi.org/10.1007/s10163-022-01453-2), recovery of valuable metals from mining wastes (https://link.springer.com/10.1007/s40831-021-00434-3), and construction and demolition waste management (https://doi.org/10.1016/j.jclepro.2020.121718), for instance.

-       Is the tungsten and cobalt also presented in the list of critical or strategic materials in other countries, such as China, Australia and Brazil?

-       How is the impact of tungsten recycling? For instance, the main uses of cobalt are batteries and steel, and such recycling will have almost nothing effect in cobalt supply chain. But what about tungsten?

-       The chapter 2 is not necessary for this paper. It must be shortened.

-       Chapter 3: is there any difference on chemical characterization among the materials sold over the years?

-       Is there recommendations for future studies?

Final decision

The manuscript may be accepted after revisions

Author Response

Please see the attached responses (pdf)

Reviewer 2 Report

The authors presented an article about “Direct recycling of WC-Co grinding chip”. The authors devotedly carried out a remarkable manuscript work. The article parts, especially the introduction part, looked good. Tables and figures were prepared very successfully. The article presentation was very successful. Congratulations for that. I think the paper is well organized and is appropriate for “Materials” journal but the paper will be ready for publication after minor revision.

·         The abstract looks good. Please include  significance numerical results.

·         Please improve the quality of Figure 5

·         The paper is well-organized. If your work is convenient for this journal’s context then there are many references from this journal. Cited sources should be primary ones. Namely, indexed area shows the power of a paper and directly your paper’s reliability. Please make regulations in this direction.

Author Response

Please see the attached responses (pdf)

Reviewer 3 Report

The paper, titled "Direct recycling of WC-Co grinding chip" evaluates a method for recovering valuable materials from grinding scrap. The article addresses a topic that is relevant from an industrial point of view and in terms of recycling and environmental protection.

Please respond to the following questions:

The authors wrote: „The combined use of an oil-free dry grinding process and the precedent and subsequent cleaning stages and a final milling, allowed obtaining second material and sintered products characterized by low contaminant level comparable to commercial powder from composition, granulometry, and morphology viewpoint.”

Nevertheless, will the reused material meet the appropriate requirements in terms of reuse and provide the expected and high quality of the product? Has the recovered material been reused to assess its value in terms of reuse? Have such verification tests been conducted? What results can be expected.

Is it possible to recycle the recovered material again and provide comparable properties? Is the recycling process only one-time? Because if one-time, perhaps in terms of cost it is simply not cost-effective.

Author Response

Please see the attached responses (pdf)

Round 2

Reviewer 1 Report

The study aimed the direct recycling of tungsten carbide with an oil-free dry grinding process. The quality of the manuscript has improved. The manuscript may be accepted.

I strongly recommend the authors to highlighted in the new version of the manuscript all changes from the old version. It is almost impossible to follow.